# Using Andersen's behavioral model of health care utilization in a decentralized program to examine the use of antenatal care in rural western Ethiopia

**Habtamu Tolera**[1,2]*, **Tegegne Gebre-Egziabher**[1], **Helmut Kloos**[3]

**1** Department of Geography and Environmental Studies, Addis Ababa University, Addis Ababa, Ethiopia,
**2** Department of Geography and Environmental Studies, Wollega University, Nekemete, Ethiopia,
**3** Department of Epidemiology and Biostatistics, University of California, San Francisco, California, United States of America

* habtol@yahoo.com

## Abstract

### Background

In Ethiopia, most women do not make the minimum number of antenatal care (ANC) visits recommended by WHO. This study modeled predisposing, enabling, need, and external environmental factors in the utilization of decentralized health facilities for ANC services in rural western Ethiopian communities.

### Methods

A community-based, cross-sectional study was conducted in Gida Ayana *Woreda* (District) among 454 women. Data were collected through structured questionnaires. Multinomial logistic regression was used to model the association between the explanatory variables and the use of recommended and fewer than recommended visits for ANC with reference to the base model, no ANC visits.

### Results

Only 15.2% of women made the recommended minimum number of ANC visits. Women with fewer than 2 children (AOR 10.7; 95% CI 3.0–8.4) were 10.7 times more likely received ANC service as recommended. Women with a delivery of 2 or more (AOR 9.7; 95% CI 3.7–5.2) home visits by health extension workers (HEWS) were 9.7 times more likely receiving minimum ANC services. Involvement in gainful activities had 4 times higher log odds of seeking recommended ANC (AOR 4.0; 95% CI 1.4–11.7). Women who experienced high fever were more likely to obtain the recommended ANC services (AOR 7.1; 95% CI 2.9–7.5). Residents of Ayana *Kebele* decentralization entity were 60% more likely to make the recommended number of visits to ANC (AOR 24.6; 95% CI 4.8–15.2).

**Data Availability Statement:** All relevant data are within the paper and its Supporting Information files.

**Funding:** This research is financed by both Addis Ababa University and Wollega University. The funders had no role in study design, data collection and analysis, decisions to publish, interpretation of the data and preparation of the manuscript for publication. Addis Ababa University supported by allocating budget for data collection. Wollega University supported the project by providing transportation service during data collection and allowing full salary during study leave for corresponding author. All funds have no grant numbers.

**Competing interests:** The authors have declared that no competing interests exist.

## Conclusions

Number of children, home visits, gainful activities, monthly income, high fever, and decentralized administrative *kebele* were strongly linked with recommended ANC schedule. The need for a program intervention aimed at meeting WHO recommendations for ANC visits include economizing birth size and spacing; improving home attendance by HEWs, knowledge of pregnancy complications and benefits of minimum ANC visits, local socio-economic development measures targeting poor women/households; further decentralization of health system improving proximity to ANC in rural western Ethiopia.

## Introduction

About 303,000 maternal deaths were reported worldwide in 2015. Of these, 99% were in the developing world, making the maternal mortality rate (MMR) in that region 239 per 100,000 live births (LBs), which was 20 times higher than in industrialized countries. Sub-Saharan African women accounted for roughly 66% (201 000) of the global maternal deaths and had the highest MMR, 546 deaths per 100,000 LBs in 2015 [1].

In Ethiopia, 13 017 maternal deaths were reported in 2015 [2]. The global burden of disease studies of 2013 and 2015 revealed MMR of 497 and 410 per 100,000 LBs, respectively, showing no significant change between the two studies [2,3]. The prevention of maternal mortality is a priority for the World Health Organization; the UN Sustainable Development Goal (SDG) and the Ethiopian government [4,5]. In the SDG period, the target is to reduce the global MMR to less than 70 per 100,000 LBs by 2030 with no country having MMR more than 140 per 100,000 LBs [5]. In 2015, the Health Sector Transformation Plan (HSTP) of Ethiopia targets for improving maternal health is to reduce MMR from 420 per 1000,000 LBs in 2015 to 199 per 100,000 in 2020 [4]. Achiveing this target by the year 2020 will also enable the country to reach her SDG3's promise of less than 140 MMR per 100,000 LBs in 2030. However, the countrywide 2016 Demographic Health Survey (EDHS) documented a MMR of 412 [6], far short of the HSTP and SDG3 targets [1,4]. Most maternal deaths (90%) are avoidable with timely interventions [7,8].

Studies carried out elsewhere have found that simple ANC interventions such as monitoring blood pressure and body weight, giving vaccinations, and providing counseling on pregnancy and danger signs are highly effective preventive measures [7,9–12]. Non-utilization of local ANC programs may help explain the persisting high rates of pregnancy complications in Ethiopia [9,13,14]. Despite the Ethiopian government's efforts to improve maternal health and bring facilities closer to mothers through decentralization programs implemented in the early 1990s [15,16,17], a recommended minimum number of ANC utilization remains low [18–20]. The 2016 EDHS, which covered all the regions of Ethiopia, found that 49.3% of women in the most populous administrative region, Oromia, did not receive ANC, and results were similar for Somali Region [6]. This statistics suggests a gap in understanding of the impact of multifaceted factors on the utilization of the minimum number of ANC visits, particularly in Oromia Region, where antenatal outcomes are poor compared to national figures [4]. Preventable maternal health risks may be managed with early detection [7,10,13,21,22]. This study examined antenatal service utilization behavior of women in a remote rural area in western Ethiopia.

Various studies have reported that ANC utilization is driven by factors such as awareness among service users and the wider communities, knowledge of maternal pregnancy and risks, community customs, previous facility use, parity and pregnancy complications, individual attitudes and health-care-seeking behaviors, household income, occupation, decision power, home visit and availability and accessibility of health facilities [13,23–26]. However, the determinants of utilization of ANC vary across different geographical locations and contexts, different cultures and beliefs, and socio-economic and demographic settings [10,12,19,25,27,28]. There is little information about women's utilization of decentralized health facilities for ANC and underlying factors in Oromo culture which has its own peculiar geographic, socio-economic, and cultural characteristics that may affect the utilization of the minimum number of ANC services. We adapted the behavioural model framework of Andersen [26] for use of health services to identify the factors potentially facilitate or impede minimum number of antenatal health services seeking behavior at individuals and community levels [13,22,29]. The model groups and predicts that a series of factors predisposing, enabling, and need and external factors influence the utilization of health services. Predisposing factors are socio-demographics characteristics; enabling factors facilitates individuals to use services such as availability of resources such as income, access to free services, availability and access to the service; need factors are physical conditions illness or disease conditions that motivate service use [26]. ANC outcomes need to be modeled as functions of these factors [13,22,29,30]. Studies employing the model in Ethiopia have used secondary sources; thus there is a need for studies using primary data [20,23,29]. Hence, the objective of this study was to investigate the above domains of determinants influencing the use of decentralized health facilities for the recommended number of ANC services with the aim of informing policy makers and practitioners responsible for planning, administering, and delivering maternal health service programs.

## Methodology

### Study setting

The study was conducted in Gida Ayana *Woreda* [15], Oromia Region, rural western Ethiopia, about 450 km from Addis Ababa, and 112 km from Nekemte, the capital of Eastern Wollega Zone. The area of the *woreda* is about 1,502 square km and organized into 7 urban and 21 rural *kebeles* (the smallest administrative units in Ethiopia). According to the 2013 population projection release and Oromia Regional State, the *woreda* had a total population of 140,484, including 47,040 child-bearing women and 10, 577 women of reproductive age, 15–49 years [31,32]. The *woreda* has 1 primary hospital, 5 health centers, and 28 health posts [32]. All promotive and preventive health servies and basic essential obstetric care are provided in the health center and health posts where as comprehensive essential obstetric care is provided in the primary hospital [33]. There were also 8 private drug shops, 3 private drug venders and 1 clinic under NGO ownership [32].

### Study design and period

A community-based, cross-sectional design was used in this study, from November 2016 to January 2017.

### Sample and recruitment

The sample size was determined using a single population proportion formula. Following a previous study [34], a proportion of 32.7%, a 95% confidence interval (CI), a margin of error of 5%, and with 2 design effect (since the selection was conducted in two stages: at *kebele* and

household level) of 1.5 were used. Thus, a minimum adequate sample size was determined using the statistical estimation method [35]. Since the source population was estimated less than 10,000; sample size correction was performed. Then, 5% non-response rate was added to obtain the adequate sample size of 459.

Two-stage sampling using a simple random sampling technique was employed to select an appropriate representative study population. In the first stage, four representative *kebeles* were randomly selected using the lottery technique. In the four selected *kebeles*, women who reported to have had their last birth during the 5 years prior to the study were identified with the help of female HEWs and women team leaders. In the second stage, eligible women were sampled using Microsoft Office Excel-generated random numbers proportional to the estimated number of women who had given birth in the respective *kebeles* during those 5 years. When two women were living in the same household, recent births were considered in determining whom to interview. Where study mothers were not available during the survey, they were visited again the next day and, if not available then, they were considered to be non-responders. Inclusion criteria of our study were women aged 15–49 years who gave birth to children during the 5 years prior to the survey. Women who reside in the respective *kebele* for a minimum of 8 months prior to data collection were not considered in this survey.

## Outcome

ANC is defined here as at least one visit to a doctor, nurse, midwife, or trained traditional birth attendant during pregnancy [36]. The nominal dependent variable of the study was the number of visits to ANC service clinics. ANC users were categorized into three groups according to the WHO recommendations for ANC visits [37], irrespective of when in the course of the pregnancy the visits occurred; the groups were as follows: the logit or the log-odds of having y = 2 [those who made the recommended 4 or more ANC visits], y = 1 [those who made 1 to 3 ANC visits, fewer than the recommended number]; and y = 0 [no ANC visits]. No ANC visit was a base model for the first two modeled categories.

## Explanatory variables

In the present study, based on Andersen's behavioral model (S1 Fig) of health care utilization theory [25,26,38], age as a three-categorical variable (19 or less, 20–34, 35–48 years), mother's and father's education (college or higher, secondary, primary, no education), marital status (married or other), mother's religion (Christian or Muslim), and decision making regarding use of household resources (husband only, husband/wife, wife only) were examined as predisposing determinant factors of adherence to the recommended number of visits to ANC clinics.

Enabling factors were mother's employment (skilled employee, small business/service, farmer, housewife), husband's employment (skilled employee, merchant, farmer, other), home visits HEWs (more than one, one, none), distance to ANC (under 30 minutes, 30 minutes or more), household income (less than 50 $US, 50 $US or more), and possession of a radio/TV (yes or no).

Need factors were severe headache, vaginal bleeding/gush, swelling of hands/face, high fever, severe pain in the abdomen, high blood pressure, and blurred vision, each classified as a binary variable (yes or no). As external environmental factors, we examined the administrative *kebeles* where the women lived and in which health facilities were decentralized as well as whether the women lived in urban or rural locations. Using these external factors enabled us to explore differences in ANC utilization across space and to consider the available decentralized health facility type (health post, health center, hospital/clinic) as an enabling factor (Fig 1).

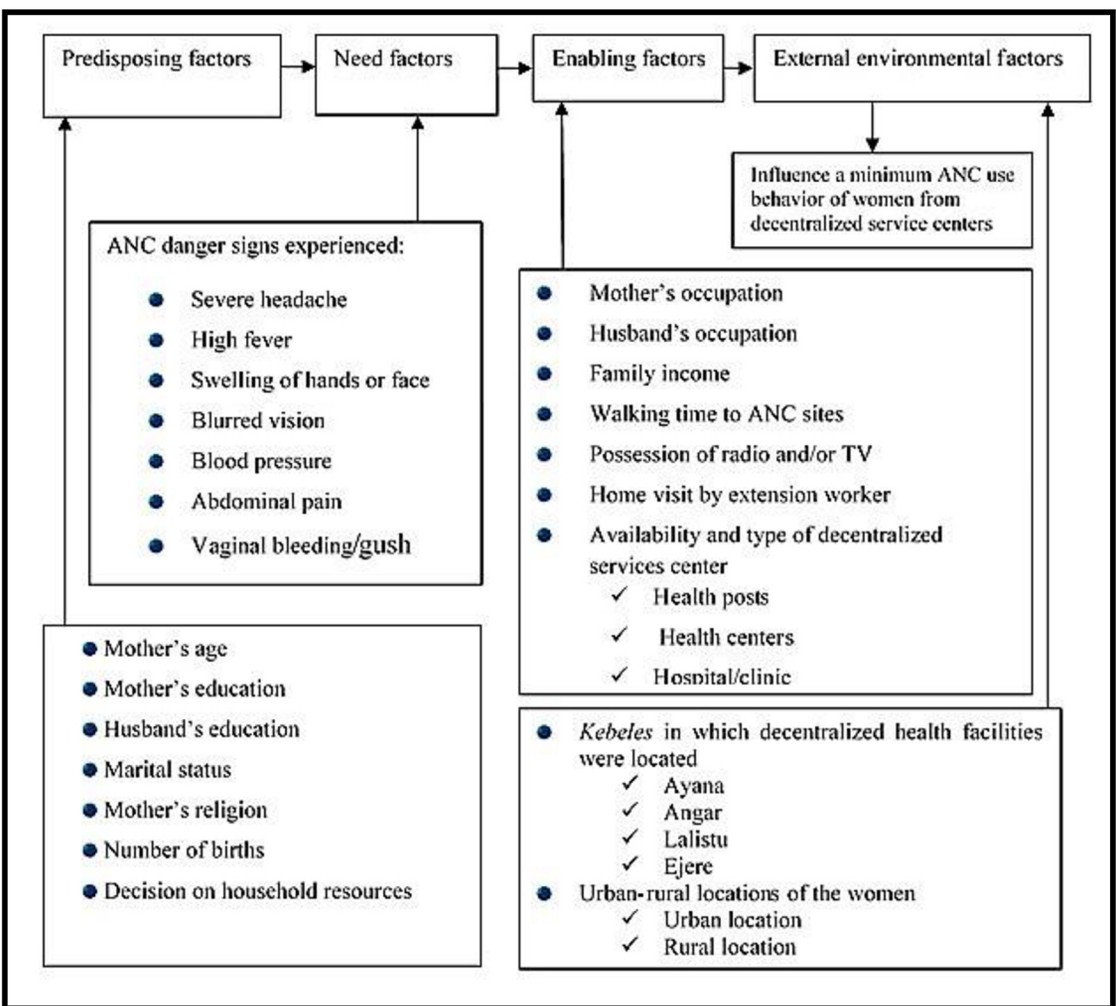

**Fig 1. Conceptual framework for health-service utilization behavior modified from Andersen's behavioral model.**

## Data collection and quality control

Data were collected using a structured questionnaire. The questionnaire was designed in English (S1 Text) and translated to Afan Oromo by staff of the English and Local Language departments of Wollega University. Data collection focused on institutional delivery; place of delivery; status of ANC visit and predisposing, enabling, perceived needs and external environmental factors affecting ANC use. Data were collected by 8 HEWs with BSc degrees or diplomas in health science who had previous experience in data collection in Gida Ayana *Woreda*. Data collection was supervised by two supervisors recruited from a local health center and coordinated by the corresponding author. The data collectors and supervisors were trained for two days in data collection techniques and ethics. A pilot study of 10% of the study population was carried out to test the survey instrument in an adjacent *woreda* (or Guto Gida) to ensure reliability, to check for clarity and comprehension. The supervisors and the principal investigator supervised and monitored data collection activities and checked all the complete questionnaires for consistency and missing data daily. Incomplete questionnaires deemed to have problems were returned to the interviewers for completion. The questionnaire was pretested

for construct validity with a 10% sample in a nearby *woreda* and modified. Each completed questionnaire was checked daily to ascertain that all the questions were correctly answered to address data validity and reliability.

## Data processing and analysis

We used statistical software EpiData Version 3.1 for preliminary data preparation and statistical software SPSS Version 24.0 for data analysis. Descriptive statistics were used to calculate the frequency distribution and proportions for categorical variables. For the normally distributed continuous variables, mean with SD was also used. The Variance Inflation Factor (VIF) > 10 indicates redundancy among explanatory variables [39,40]. Our ANC utilization model satisfied this criterion with VIF < 2.0. Associations between the number of ANC service visits in the three groups (those receiving the recommended number of ANC visits, those receiving fewer than recommended, and those receiving no ANC) and explanatory variables were calculated by the use of the binomial and multinomial logit (MNL) model.

The associated factors were examined using chi-square test and multivariable logistic regression analysis. All the significant variables in the bivariate analysis ($p < 0.05$) were included in the the final multinomial logistic (MNL) regression model because bivariate association between two variables does not necessarily imply a significant causal relationship between them. Therefore, a multivariate approach was applied to determine which factors best explain and predict health care service use outcome. The adequacy of the developed model was verified through the standard statistical mean of likelihood ratio test of goodness of fit [40]. Multicollinearity in the MNL model was detected by examining the standard error for the coefficients [11]. Adjusted odds ratios (AOR) with corresponding 95% CI estimates were used to describe the strength of associations of factors with recommended number of ANC visits and fewer than the recommended number of ANC visits versus no visits. The association of variables was found to be statistically significant at $p < 0.05$.

## Ethical consideration

The research protocol was reviewed and approved by Wollega University Research Ethics Approval Committee [Ref/No.WU–99529/H1-3/24/11/2016]. Permission was received from Gida Ayana *Woreda* Health Office. The purpose of the study was explained to all participants and a consent form approved by the Review Board was given to participants. Parents or legal guardians provided consent for all participants under age 18. We emphasized that participation was completely voluntary and that they had the right to withdraw any time during the interview without giving any reason. Confidentiality and anonymity were explained to all participants. We ensured that all participants understood the information given by asking them. The consent form was read aloud for women who could not read or write. Literate women were asked to read the consent form. A written consent in the form of a signature or a thumbprint was obtained from all of the participants.

## Results

### Socio-demographic characteristics and nature of home visit

A total of 459 women who had their last birth during the 5 years preceding the survey were enrolled in the study. The response rate was 98.9%. The mean age of mothers was 26.1 (±7.1) years. The mean number of children women gave birth to was 3.11 (±2.0). Nearly half of the mothers reportedly had no formal education. Over one-third of the husbands had completed secondary education and 33.9% completed college/higher education. Forty-nine percent of the

respondents were of the Oromo ethnic group, 83.9% were married, and 55.9% were urban. Over half of mothers were Christian. One hundred ninety-eight (43.6%) were housewives and 124 (27.5%) were paid employees. Most husbands (46.7%) were subsistence farmers. Mean walking time it takes pregnant women to reach the nearest health facility was 51.1(±31) minutes, and the mean monthly household income was 47 (±15.1) $US. The majority of participants (56.2%) reported that they did not obtain home visit and support by HEWs (Table 1).

## ANC service use and decentralized facility attended

The majority (55.1%) of the women obtained ANC from a decentralized health center facility. The women who made the recommended number of visits to ANC constituted 15.2% of the participants while 49.6% made fewer than the recommended number and 35.2% did not seek services. We found that 89.4% and 66.2% of respondents of Ayana and Ejere *Kebeles*, respectively, visited the ANC clinic, and lower proportions did so in Angar and Lalistu *Kebeles*. Of the women who utilized ANC services, 41.5% made their first visit in the second trimester of their pregnancy. Among those who made no ANC visit, 46.2%, 25.0%, and 11.2% mentioned lack of awareness about the importance of pregnancy care, transportation problems, and long waiting times, respectively, as reasons for not using ANC services (Table 2).

## Factors influencing visits to ANC in a decentralized facility

The results of bivariate analyses of number of ANC visits and the independent variables showed that walking distance to neaby facility, sever headache, vaginal bleeding and rural urban residence appeared to be positively associated with minimum number of recommended ANC received ($p < 0.05$). Maternal age at last pregnancy, mother's and husband's education, marital status, religion, number of births, decision on family resources, type of decentralized facility, monthly income, occupation, home visit by HEWs, availability of radio/orTV, high fevere, swelling of hands/face, abdominal pain and *Kebele* in which decentralized health facilities were located were also significantly associated ($p < 0.01$) with the recommended schedule of ANC visits. No statistically significant associations were found for high blood pressure and blurred vision assessed ($p > 0.05$).

In the following sections, we present the influence of each determinant factor on recommended number of ANC visits and fewer than the recommended ANC visits versus no ANC visits as established through the multinomial regression analysis and shown in Table 3.

**Predisposing factors.** Holding other variables constant, Christian women were 3.3 times more likely than Muslims to make 3 or fewer ANC visits than no visits. Nevertheless, identification as Christian was not significantly associated with utilization of the recommended number of ANC visits although the corresponding log odds figure was higher (AOR = 1.6; 95% CI 0.6–3.7, $p > 0.05$).

Women who had fewer than 2 children were 10.7 times and 9.2 times more likely to make the recommended number of ANC visits (AOR = 10.7; 95% CI 3.0–8.4, $p < 0.01$) and fewer than the recommended number (AOR = 9.2; 95% CI 3.6–23.0, $p < 0.01$), respectively, than no visits compared to women who reported 4 children or more. Likewise, women with 2 to 3 children had log odds of 5.5 times and 4.6 times higher of making the recommended number of ANC visits (AOR = 5.5; 95% CI 1.5–2.4, $p < 0.05$) and fewer than the recommended ANC visits (AOR = 4.6; 95% CI 1.8–11.5, $p < 0.01$), respectively, than no visits compared to women who had 4 or more children.

Gendered decision making about resource use by husband (AOR = 1.5; 95% CI 0.6–3.9, $p > 0.05$) and wife/husband (AOR = 1.6; 95% CI 0.6–4.6, $p > 0.05$) influenced women's maternal health service seeking behavior in choosing the recommended number of ANC visits

**Table 1. Socio-demographic backgrounds of the study participants and home visits, in Gida Ayana *Woreda*, rural western Ethiopia, during Nov. 2016 to Jan 2017 ($N$ = 454).**

| Variable | Variable categories | Number($n$) | Percentage (%) |
|---|---|---|---|
| Age (in years) | 19 or less | 127 | 28.0 |
| | 20–34 | 255 | 56.1 |
| | 35 or more | 72 | 15.9 |
| Education | No formal education | 224 | 49.3 |
| | Primary | 81 | 17.8 |
| | Secondary | 73 | 16.1 |
| | College/higher | 76 | 16.7 |
| Husband's education | No formal education | 90 | 19.8 |
| | Primary | 49 | 10.8 |
| | Secondary | 161 | 35.5 |
| | College/higher | 154 | 33.9 |
| Occupation | Paid employee | 124 | 27.5 |
| | Small business/service | 98 | 21.6 |
| | Farmer | 34 | 7.3 |
| | Housewife | 198 | 43.6 |
| Husband's occupation | Skilled employee | 86 | 18.9 |
| | Merchant | 114 | 25.1 |
| | [a]Informal activity | 42 | 9.3 |
| | Farmer | 212 | 46.7 |
| Marital status | Married | 381 | 83.9 |
| | [b]Other | 73 | 16.1 |
| Ethnicity | Oromo | 222 | 48.9 |
| | Amhara | 144 | 31.7 |
| | Tigre | 88 | 19.4 |
| Religion | Christian | 260 | 57.3 |
| | Muslim | 194 | 42.7 |
| Urban-rural location of the women | Urban | 254 | 55.9 |
| | Rural | 200 | 44.1 |
| Mean number of births (SD) | 3.11(±1.9) | | |
| Mean walking time to ANC clinic in min (SD) | 51.1(±30.9) | | |
| Mean monthly family income in $US (SD) | [c]47.0(±15.1) | | |
| Home visit by HEWs | 2 times or more | 132 | 29.0 |
| | One time | 67 | 14.8 |
| | No | 255 | 56.2 |

[a]Informal activity: day laborer/weaving/students.

[b]Other: single/widowed/divorced. SD: Standard Deviation.

[c]Average exchange rate of 1$US was 21.43 Ethiopian Birr between November 2016-January 2017.

rather than no visits, but the association was not statistically significant. However, when all other factors were held constant, shared decision making (wife/husband) about household resource use was 3.9 times more likely to result in fewer than the recommended ANC clinic visits than no visits compared to wife-only decision making (AOR = 3.9; 95% CI 1.8–8.4, $p$ < 0.01).

**Enabling factors.** When all other determining factors were held constant, women who operated small businesses had 4.0 times (AOR = 4.0; 95% CI 1.4–11.7, $p$ < 0.01). and 2.2 times (AOR = 2.2; 95% CI 1.0–4.8, $p$ < 0.01) higher log odds of choosing recommended and fewer

**Table 2. Utilization of ANC and local health facilities attended, in Gida Ayana *Woreda*, rural western Ethiopia, Nov. 2016-Jan. 2017 (N = 454).**

| Variable | Variable categories | Number(n) | Percentage(%) |
|---|---|---|---|
| ANC visits | Recommended # ANC visits | 69 | 15.2 |
| | Fewer than recommended visits | 225 | 49.6 |
| | No visits | 160 | 35.2 |
| Decentralized facility visited | Hospital/clinic | 25 | 8.5 |
| | Health center | 162 | 55.1 |
| | Health post | 107 | 36.4 |
| Decentralized administrative *kebele* by ANC visits | Ayana | 84 | 89.4 |
| | Ejere | 51 | 66.2 |
| | Angar | 92 | 57.5 |
| | Lalistu | 67 | 54.5 |
| Timing of 1$^{st}$ ANC visit | 1$^{st}$ trimester | 96 | 32.6 |
| | 2$^{nd}$ trimester | 122 | 41.5 |
| | 3$^{rd}$ or 4$^{th}$ trimesters | 76 | 25.9 |
| [c]Reason for no ANC visit | Lack of awareness | 74 | 46.25 |
| | Transportation problem | 40 | 25.0 |
| | Waiting time | 18 | 11.25 |
| | Illness was not severe | 14 | 8.75 |
| | Heavy workload | 10 | 6.25 |
| | Others | 4 | 2.5 |
| | Total | 160 | 100 |

[c]multiple responses were possible.

than the recommended number of ANC visits, respectively, than no visits compared to women who identified as housewives.

Women in households with monthly household income of 50 $US or more were nearly 3 times and 2 times more likely to make the recommended (AOR = 2.8; 95% CI 1.2–6.2, $p < 0.05$) and fewer than recommended number of ANC visits (AOR = 2.1; 95% CI 1.1–3.8, $p < 0.05$), respectively, than no visits compared to those reporting less than 50 $US in household income.

Women who were visited 2 times or more than 2 times by HEWs were 9.7 times and 4.2 times, respectively, more likely to make the recommended (AOR = 9.7; 95% CI 3.7–5.2, $p < 0.01$) and fewer than recommended number of ANC visits (AOR = 4.2; 95% CI 1.9–8.9, $p < 0.01$) than no visits compared to those reporting no visits by HEWs. Likewise, women who were visited one time had higher log odds of making the recommended (AOR = 9.5; 95% CI 2.9–3.7, $p < 0.01$) and fewer than recommended (AOR = 4.8; 95% CI 1.8–13.0, $p < 0.01$) number of ANC visits by 50% and 80%, respectively, than no visits compared to those who were not visited by HEWs.

Walking time strongly influenced the utilization of health facilities, specifically the number of ANC visits, but the relationship was not statistically significant (AOR = 1.7; 95% CI 0.7–4.4, $p > 0.05$). Furthermore, the log odds ratio of receiving fewer than 4 ANC visits versus no visits was 40% higher for women living closer than 30 minutes from the nearest ANC clinics (AOR = 2.4; 95% CI 1.2–5.0, $p < 0.05$).

When comparing decentralized health facility types of hospital/clinic (AOR = 1.1; 95% CI 0.2–4.6, $p > 0.05$). and health center (AOR = 2.3; 95% CI 0.9–5.8, $p > 0.05$), no significant association was found with the utilization of the recommended number of ANC visits versus

**Table 3. Multinomial regression analysis for factors influencing number of women's ANC visits, in Gida Ayana *Woreda*, rural western Ethiopia, Nov. 2016-Jan. 2017 (*N* = 454).**

| Factor | Variable | Variable categories | Status of ANC visits. Base model: No-visits category | | | |
|---|---|---|---|---|---|---|
| | | | Recommended visits | | Fewer than recommended visits | |
| | | | AOR (95% CI) | *p*-value | AOR (95% CI) | *p-value* |
| Predisposing factors | Religion | Christian | 1.6(0.6–3.7) | 0.279 | 3.3(1.7–6.5)** | 0.001 |
| | | Muslims | 1 | | 1 | |
| | Marital status | Married | 0.5(0.1–1.5) | 0.253 | 2.3 (0.9–5.8) | 0.06 |
| | | others | 1 | | 1 | |
| | No.of children | Fewer than 2 | 10.7(3.0–8.4)** | 0.001 | 9.2(3.6–23.0)** | 0.001 |
| | | 2 to 3 | 5.5(1.5–2.4)* | 0.01 | 4.6(1.8–11.5)** | 0.001 |
| | | 4 or more | 1 | | 1 | |
| | Age | 19 or less | 1.7(0.4–6.5) | 0.402 | 1.0(0.4–2.7) | 0.906 |
| | | 20–34 | 0.6(0.2–2.3) | 0.557 | 0.9(0.3–2.1) | 0.814 |
| | | 35 or more | 1 | | 1 | |
| | Education | College/higher | 1.1(0.2–5.1) | 0.902 | 0.7(0.2–2.7) | 0.719 |
| | | Secondary | 0.8(0.2–2.7) | 0.743 | 0.6(0.2–1.7) | 0.395 |
| | | Primary | 1.7(0.5–6.0) | 0.373 | 2.1(0.8–5.1) | 0.092 |
| | | No education | 1 | | 1 | |
| | Husband's education | College/higher | 2.4(0.4–12.7) | 0.299 | 0.8(0.2–3.1) | 0.771 |
| | | Secondary | 1.7(0.3–7.7) | 0.468 | 1.3(0.4–4.1) | 0.64 |
| | | Primary | 1.2(0.4–3.9) | 0.662 | 0.9(0.4–2.0) | 0.878 |
| | | No education | 1 | | 1 | |
| | Decision on family resource | Husband | 1.5(0.6–3.9) | 0.334 | 1.9(0.9–4.0) | 0.075 |
| | | Wife/husband | 1.6(0.6–4.6) | 0.316 | 3.9(1.8–8.4)** | 0.001 |
| | | Wife | 1 | | 1 | |
| Enabling factors | Employment | Employee | 2.1(0.7–5.8) | 0.133 | 1.4(0.6–2.9) | 0.365 |
| | | Small business/service | 4.0(1.4–11.7)** | 0.009 | 2.2(1.0–4.8)* | 0.05 |
| | | Farmer | 2.3(0.4–13.4) | 0.349 | 1.8(0.3–9.0) | 0.424 |
| | | Housewife | 1 | | | |
| | Husband's employment | Employee | 1.5(0.2–8.1) | 0.623 | 2.0(0.5–7.7) | 0.291 |
| | | Merchant | 1.0(0.3–3.1) | 0.972 | 1.5 (0.6–3.5) | 0.341 |
| | | Other | 0.8 (0.1–4.0) | 0.885 | 1.3(0.4–3.9) | 0.603 |
| | | Farmer | 1 | | 1 | |
| | Family income | 50 $US or more | 2.8(1.2–6.2)* | 0.011 | 2.1(1.1–3.8)* | 0.013 |
| | | Less than 50 $US | 1 | | 1 | |
| | Home visit by HEWs | More than one time | 9.7(3.7–5.2)** | 0.001 | 4.2(1.9–8.9)** | 0.0001 |
| | | One time | 9.5(2.9–3.7)** | 0.001 | 4.8(1.8–13.0)** | 0.001 |
| | | No visit | 1 | | 1 | |
| | Walking time to ANC clinic | Less than 30 minutes | 1.7(0.7–4.4) | 0.226 | 2.4(1.2–5.0)* | 0.013 |
| | | 30 minutes or more | 1 | | 1 | |
| | Possession of radio/or TV | Yes | 1.7(0.7–4.1) | 0.224 | 1.7(0.8–3.3) | 0.108 |
| | | No | 1 | | 1 | |
| | Decentralized facilities | Hospital/clinic | 1.1(0.2–4.6) | 0.889 | 0.4(0.1–1.6) | 0.239 |
| | | Health center | 2.3(0.9–5.8) | 0.08 | 2.2(1.1–4.5)* | 0.019 |
| | | Health post | 1 | | 1 | |

(*Continued*)

**Table 3.** (Continued)

| Factor | Variable | Variable categories | Status of ANC visits. Base model: No-visits category | | | |
| | | | Recommended visits | | Fewer than recommended visits | |
| | | | AOR (95% CI) | *p*-value | AOR (95% CI) | *p-value* |
|---|---|---|---|---|---|---|
| Need factors | Severe headache | Yes | 2.9(1.1–7.5)* | 0.026 | 3.7(1.8–7.6)** | 0.001 |
| | | No | 1 | | 1 | |
| | Vaginal bleeding/gush | Yes | 1.7(0.6–4.6) | 0.239 | 2.2(1.1–4.8)* | 0.036 |
| | | No | 1 | | 1 | |
| | Swelling of hands/face | Yes | 1.2(0.5–2.9) | 0.601 | 0.9(0.5–1.8) | 0.99 |
| | | No | 1 | | 1 | |
| | High fever | Yes | 7.1(2.9–7.5)** | 0.001 | 4.1(1.9–8.5)** | 0.001 |
| | | No | 1 | | 1 | |
| | Severe pain in abdomen | Yes | 0.8(0.3–2.4) | 0.794 | 1.1(0.5–2.5) | 0.676 |
| | | No | 1 | | 1 | |
| External Environmental factors | *Kebeles* in which decentralized health facilities were located | Ayana | 24.6(4.8–15.2)** | 0.001 | 8.2(2.1–3.5)** | 0.002 |
| | | Angar | 2.0(0.3–11.3) | 0.398 | 0.9(0.2–3.3) | 0.951 |
| | | Lalistu | 1.2(0.3–5.1) | 0.742 | 0.8(0.3–2.5) | 0.806 |
| | | Ejere | 1 | | 1 | |
| | Urban-rural residence | Urban | 2.1(1.1–3.7)** | 0.013 | 1.4(0.9–2.1) | 0.8 |
| | | Rural | 1 | | 1 | |

*1* = Reference category.

** significant at $p < 0.01$

*significant at $p < 0.05$

no visits. For women who received services at decentralized front-line health posts, the log odds of making fewer than the recommended number of ANC visits versus no visits to health centers was 20.0% higher (AOR = 2.2; 95% CI 1.1–4.5, $p < 0.05$).

**Need factors.** When all other determining factors were held constant, women who reportedly felt severe headaches were found to be 90% (AOR = 2.9; 95% CI 1.1–7.5, $p < 0.05$) and 70% (AOR = 3.7; 95% CI 1.8–7.6, $p < 0.01$) more likely to make the recommended and fewer than recommended number of ANC visits, respectively, than those not reporting headaches. The log odds ratio for making fewer than the recommended number of ANC visits was 20% higher (AOR = 2.2; 95% CI 1.1–4.8, $p < 0.05$) for women who experienced vaginal bleeding/gush compared to those who did not. Likewise, the odds of women who had high fevers were 7.1 and 4.1 times higher for having the recommended (AOR = 7.1; 95% CI 2.9–7.5, $p < 0.01$) and fewer than recommended (AOR = 4.1; 95% CI 1.9–8.5, $p < 0.01$) number of visits to ANC facilities, respectively, than those who felt no fever.

**External environmental factors.** When comparing the utilization of ANC services in the four study *kebeles* in which government decentralized health facilities were located, women of Ayana had higher odds of making the recommended number of visits (AOR = 24.6; 95% CI 4.8–15.2, $p < 0.01$) and fewer than the recommended number (AOR = 8.2; 95% CI 2.1–3.5, $p < 0.01$) than making no visits compared to women of Ejere *Kebele*. Furthermore, when all other factors were held constant, residents of urban settlements were 2.1 times more likely to make the recommended number of ANC service visits (AOR = 2.1; 95% CI 1.1–3.7, $p < 0.01$) compared to rural residents. However, the urban-rural difference did not seem to significantly influence the choice of fewer than the recommended number of ANC service visits in spite of higher log odds(OR = 1.4; 95% CI 0.9–2.1, $p > 0.05$).

## Discussion

Although increasing the number of ANC visits has contributed to a drastic reduction in the maternal death rate in low-income countries during the past 30 years, the majority of women in sub-Saharan Africa, including Ethiopia, still do not make the WHO-recommended 4 ANC visits or more during the pregnancy period [21]. This study found that only 15.2% of the 454 participants received the recommended number of ANC visits; 49.6% made fewer than the recommended ANC visits and 35.2% reported no visits for ANC.

Our findings identified a number of predisposing, enabling, need, and environmental factors influencing the choices regarding ANC visits: religion, number of children, woman's occupation, home visit by HEWs, walking time to health facility, monthly income, severe headache, vaginal bleeding/guish, high fever, availability of decentralized ANC facilities, decentralized administrative *kebele*, and urban-vs-rural residence of the women.

After adjusting for all variables, Christian women were found to be 30% more likely to make fewer than the recommended minimum number of ANC visits than no visits compared to Muslim women although religion was not significantly associated with making the recommended number of ANC visits. This finding is consistent with a study in northeastern Ethiopia, where Christians were 2.2 times more likely to make fewer than the recommended number of ANC visits compared to Muslim women [41]. In Nigeria, Christian women were more likely to make the recommended number of ANC visits than fewer than the recommended number than Muslims [42]. The higher level of ANC use among women of certain religions could be attributed to theological differences and differences in lifestyle across various beliefs [12]. In Nepal, Christians and Hindus were 50% and 30%, respectively, more likely to make the recommended and fewer than recommended number of ANC visits versus no visits compared to women of other religions [11].

Women experiencing their first pregnancy and those who had 2 or 3 children were 10.7 and 5.0 times more likely to make the recommended minimum number of ANC visits, respectively, than no visits compared to baseline. Recent systematic reviews and meta-analyses reveal that women with first pregnancies are more likely than multiparous women to make the recommended number of ANC visits due to fear of complications with the first birth [21,22,29]. Multiparous women tend to believe there is less risk to current pregnancy due to their previous birthing experiences and their negative perceptions of the environment in local health institutions regarding cleanliness, equipment quality, and behavior of providers [13,21,27,43].

Decision making status on family resourece was associated with increased log odds of utilizing minimum antenatal care services among mother. Our results also showed that women who were able to decide with their partners on family resource use had higher log odds of choosing fewer than the recommended number of ANC visits by 90.0%, which implies that housewives with some autonomy in this area were able to make at least some visits to clinics. This suggests that women who were not constrained by a patriarchal structure were better able to utilize ANC services. This finding corroborates a study in northern Ethiopia [18] in which decision making by wives and husbands separately was associated with 45% and 65% lower numbers making the recommended and fewer than recommended number of ANC visits, respectively, compared to couples who made the decision jointly.

Amongst the enabling factors, the odds for utilizing minimum number of ANC clinics increased among mothers who were engaged in non-housewife types of occupations. Our data showed that the log odds ratios associated with gainful small business activities were 4.0 times higher for making the recommended minimoum number of ANC visits. The odds ratios corresponding to the other categories of women's occupation were also higher. However, their *p*-values did not demonstrate statistically significant association with ANC service visits.

Engaging in skilled employement and small businesses as income sources among mother was associated with increased odds of utilizing recommended minimum number of ANC services which is consistent with those of previous studies in Ethiopia [21,24]. A study in Nigeria [44] reported that women who operated small businesses were 6 times more likely to make the recommended number of ANC clinics visits. Similar studies in Nepal and China also corroborate our findings [11,45].

Furthermore, we found that women in households reporting monthly income of $US 50.00 or more made the recommended minimum number of ANC visits and fewer than recommended visits at a rate 80% and 10% higher than no visits, respectively, compared to those with household incomes of $US 49 or less. Women from high household incomes were more likely to be able to afford health services, and their associated costs, including transportation costs [46,47]. Low household income denoted a major deterrent to mothers to seek prompt care. This variation might be an area of concern for policy makers. A study in Sodo *Woreda* [21] reported that 13% of rural women with higher cash incomes made the ANC visits recommended by WHO. Similar findings were reported in Nigeria, where women in wealthier households were 4.0 times more likely to make the recommended minimum number of ANC visits [48]. The differentials were 2.71 times in studies in Afghanistan [49], 8.8 times in Nepal [11], and 3.3 times in China [45].

Women who were visited 2 or more times by HEWS were 70% and 20% more likely to make the recommended and fewer than recommended number of visits, respectively, than women who were not visited. Women who were visited only once made the recommended minimum ANC visits and fewer than recommended number of visits at a 50% and 80%, respectively, higher rate. A qualitative study in Kafa Zone, southwestern Ethiopia [50] reported that women preferred to be seen by HEWs who they knew rather than health workers they did not know. Similar oversea findings were reported by several other studies [12,49,51,52]. This finding may be important to develop intervention strategies that more strengethened a systematic and regular home visits by health workers which helps women to improve their utilization of a minimum ANC services or more at home or health post with low opportunity costs.

Women who visited decentralized health centers were 20% more likely to make 3 or fewer ANC visits than those with utilized nearby bottom-line health posts. This finding corroborates those of other studies, which predicted higher log odds for making at least 3 recommended visits to ANC clinics [51,53]. Women were likely to visit health centers that were better equipped and more user friendly than others [54]. But this result was not consistent with the studies in Kaffa Zone, southwestern Ethiopia where majority of study participants reported that they used a minimum number of ANC at the health post rather than the health centre because of the physical distance, the cost of health services at health centres or the hospital and because women preferred to be seen by HEWs who they knew rather than health workers they did not know [50], and in Rwanda, where decentralized health posts were over utilized compared to health centers because health posts were better supplied with maternal resources and attracted most of the local women [55].

After adjusting for all variables, women were found to be 40% more likely to make fewer than recommended minumum number of ANC visits to health facilities located within walking distance of half an hour than to facilities located at greater distances. Similar extensive studies elsewhere [18,24,56] pointed out that walking distance to the available health facilities and time needed to reach these health facilities influences health-seeking behavior and was associated with the utilization of a minimum number of health services. An extensive study in Ethiopia reported that utilization of health facilities declined with distance from maternal service users homes [57].

Severity of pregnancy complication or illness also increased seeking care in health facilities and associated with the utilization of the minimum number of ANC visits [47]. Amongst the need factors, our data show that recognizing the severity of illness by danger signs of severe headaches increases the likelihood of utilizing minimum ANC visits and women with severe headaches had 9.0 times higher log odds of making the recommended number of visits for ANC; headaches appeared to motivate mothers to attend the minimum number of ANC clinics, similar to other studies [18,21,30,43,58]. Furthermore, an increased odds of utilizing minimum ANC services was observed in mothers of high severity of illness by danger signs of high fevere. The study found that women reporting high fevers were 7.1 times times more likely to make the recommended minimum number of ANC visits. This finding is in line with studies in Hadiya Zone in southern Ethiopia [59], west Bengal in India [60], and rural Bangladesh [43]. The reason women with fevers appear motivated to utilize ANC could be that mothers with a history of complications have personal experience that helps them understand the life-threatening condition and makes them inclined to seek preventive maternal care [21].

Log odds of attending recommended minimum number of visits to ANC were 60% among women of Ayana *kebele*, *woreda's* capital, than outer administrative decentralisation units in the rural areas. There were geographical variations in the use of ANC among women across *kebeles* of different socio-cultural groups [13] with government-decentralized health facilities. Another study found wide interregional disparities in ANC use in Ethiopia, with Oromia Region having the lowest use of all regions except Somali [6]. In India, except for the southern region, as well as in Pakistan, Nigeria, and South Sudan, ANC utilization rates are low [30,48,49]. Low ANC utilization in these areas may be due to historical, socioeconomic, and cultural conditions across these physical settings and community groups.

The log odds of urban residence of women was significantly associated with adherence to the utilization of minimum number of ANC visits than rural women by 10% higher rate. This finding is consistent with meta-analysis in Ethiopia [61]. In Nepal, urban women were 7 and 2 times more likely to make the recommended and fewer than recommended number of ANC visits, respectively [11]. Researchers suggest the reasons for the higher use might be the better quality of care and greater accessibility to professionals in urban areas. Non-significant association between urban residence and fewer than recommended number of ANC visits has been reported elsewhere [27,48].

Our study had a number of limitations. All data were self-reported by the women participants and were not triangulated with other sources, which may have resulted in bias. The study also did not capture institution-based factors such as health providers' behavior and accessibility and quality of services, all of which influence the health-seeking behavior of women. Moreover, the long recall period may have introduced information bias.

## Conclusion

Despite the fact that ANC attendance impacts safe motherhood and reduces maternal deaths, this study found unacceptably low adherence to the recommended number of ANC visits in rural western Ethiopia. Only 15.2% of women studied made the recommended number of visits to ANC clinics, 49.6% made fewer than the recommended number, and 35.2% did not obtain any ANC. The results of this study confirm the importance of several factors in women's making the WHO-recommended number of visits to ANC clinics: number of births, occupation, visits by HEWs, household income, headache, fever, decentralized administrative *kebele*, and urban-rural location. Religion, gendered decisions about resource use, walking time to ANC services, decentralized facility type, and vaginal bleeding were significant predictors of fewer than the recommended number of ANC visits.

These results indicate that the study of broadly-based interventions considering the socio-economic, cultural, demographic, and environmental context of communities may be useful in identifying barriers to ANC utilization and promoting adherence to the recommended number of maternal visits in rural western Ethiopia. Health campaigns conducted through *kebele* and *woreda* health services as well as mass media may promote health-seeking behavior of pregnant women and increase the awareness of communities, religious leaders, and other stakeholders about the recommended number of ANC visits so as to reduce maternal and neonatal mortality. However, policies and programs must extend beyond community awareness of the need for adequate and appropriate maternal care, use of family planning to control birth size and spacing, and address also long-term multi-sectoral development issues. Broadly based interventions need to focus on motorized rural roads, public transport, livelihoods and income generation, and gender equity. Interventions must increase the number and the coverage of home visits by HEWs and upgrade and equip front-line health posts or further extend the decentralization of health centers in rural areas. These actions will ensure that the predisposing, enabling, need, and external environmental factors that promote health-seeking behavior are in place to achieve increased ANC utilization and reduction in maternal deaths.

## Supporting information

**S1 Fig. Origonal Andersen's behavioural health care seeking framework.**
(DOCX)

**S1 Text. English language survey questionnaire developed to study the utilization of decentralized health facilities for minimum number ANC visits, rural western Ethiopia.**
(DOCX)

## Acknowledgments

We thank Wollega University for providing us the ethical clearance to undertake the study. We also thank Gida Ayana *Woreda* health officials and the *kebele* administrators for their cooperation during data collection. we would like to thank all of the participants of this study for their time and patience in responding to our interviews. Our special thanks also go to the data collectors and supervisors. We also want to thank Mrs. Ann Byers for editing the manuscript.

## Author Contributions

**Conceptualization:** Habtamu Tolera, Tegegne Gebre-Egziabher, Helmut Kloos.

**Data curation:** Habtamu Tolera.

**Formal analysis:** Habtamu Tolera, Tegegne Gebre-Egziabher, Helmut Kloos.

**Funding acquisition:** Habtamu Tolera.

**Investigation:** Habtamu Tolera, Tegegne Gebre-Egziabher, Helmut Kloos.

**Methodology:** Habtamu Tolera, Tegegne Gebre-Egziabher, Helmut Kloos.

**Project administration:** Habtamu Tolera.

**Resources:** Habtamu Tolera.

**Software:** Habtamu Tolera.

**Supervision:** Habtamu Tolera, Tegegne Gebre-Egziabher, Helmut Kloos.

**Validation:** Habtamu Tolera, Tegegne Gebre-Egziabher, Helmut Kloos.

**Visualization:** Habtamu Tolera, Tegegne Gebre-Egziabher, Helmut Kloos.

**Writing – original draft:** Habtamu Tolera.

**Writing – review & editing:** Habtamu Tolera, Tegegne Gebre-Egziabher, Helmut Kloos.

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
