## [Decision Letter · Decision Letter 0]

11 Dec 2019

PONE-D-19-25468

Using Andersen’s behavioral model of health care utilization in a decentralized program to examine the use of ANC in rural western Ethiopia

PLOS ONE

Dear Mr. Tolera,

Thank you for submitting your manuscript to PLOS ONE. After careful consideration, we feel that it has merit but does not fully meet PLOS ONE’s publication criteria as it currently stands. Therefore, we invite you to submit a revised version of the manuscript that addresses the points raised during the review process.

We would appreciate receiving your revised manuscript by Jan 25 2020 11:59PM. To enhance the reproducibility of your results, we recommend that if applicable you deposit your laboratory protocols in protocols.io, where a protocol can be assigned its own identifier (DOI) such that it can be cited independently in the future. For instructions see: http://journals.plos.org/plosone/s/submission-guidelines#loc-laboratory-protocols

We look forward to receiving your revised manuscript.

Kind regards,

Kannan Navaneetham

Academic Editor

PLOS ONE

Journal Requirements:

Please ensure that your manuscript meets PLOS ONE's style requirements, including those for file naming. The PLOS ONE style templates can be found at http://www.plosone.org/attachments/PLOSOne_formatting_sample_main_body.pdf and http://www.plosone.org/attachments/PLOSOne_formatting_sample_title_authors_affiliations.pdf Please ensure that you refer to Figure 2 in your text as, if accepted, production will need this reference to link the reader to the figure. Please include additional information regarding the survey or questionnaire used in the study and ensure that you have provided sufficient details that others could replicate the analyses. For instance, if you developed a questionnaire as part of this study and it is not under a copyright more restrictive than CC-BY, please include a copy, in both the original language and English, as Supporting Information.  If the original language is written in non-Latin characters, for example Amharic, Chinese, or Korean, please use a file format that ensures these characters are visible. Please state whether you validated the questionnaire prior to testing on study participants. Please provide details regarding the validation group within the methods section. Please change your reference to "p=0.000" to "p<0.001" or as similarly appropriate, as p values cannot equal zero.

Reviewers' comments:

Reviewer's Responses to Questions

**Comments to the Author**

1. Is the manuscript technically sound, and do the data support the conclusions?

Reviewer #1: Yes

Reviewer #2: Yes

2. Has the statistical analysis been performed appropriately and rigorously? 

Reviewer #1: Yes

Reviewer #2: Yes

3. Have the authors made all data underlying the findings in their manuscript fully available?

Reviewer #1: Yes

Reviewer #2: Yes

4. Is the manuscript presented in an intelligible fashion and written in standard English?

Reviewer #1: Yes

Reviewer #2: Yes

5. Review Comments to the Author

Reviewer #1: Title:

• Please avoid the use of abbreviations like (ANC) in the title.

Abstract:

• In the result part, it is better to describe the magnitude (15.2%) and then the factors. “Adherence to the recommended number of ANC service visits was a function of predisposing, enabling, need, and external environmental factors. Women who made the recommended number of ANC visits constituted 15.2% of all subjects.”

• Does this statement “Women’s involvement in gainful activities had higher log odds of seeking the recommended ANC services compared to housewives.” based on the logistic regression model result? If so, you should write the Odds ratio form the regression.

Introduction

• The statistics reported in the first paragraph, “About 300,000 maternal deaths were reported worldwide in 2013”, is rather reflected in the 2015 WHO maternal mortality report. Would you please double-check the figure?

• Paragraph two: “The Ethiopian Demographic 45 Health Surveys of 2005 and 2011 revealed maternal death ratios of 673 and 676 per 100,000 live 46 births, respectively, showing no change between the two studies.” The maternal mortality definition used by the DHS is different from the formal WHO definition used as it considers all pregnancy-related deaths. Hence using the figure from the DHS might not be correct in this context.

• Please summarise the last three paragraphs of the introduction in one succinct sentence that addresses the justification and aim of the study.

Method

• Study setting: Add some brief information regarding the health infrastructure of the study area.

• You don’t need to mention the name of the kebeles “(Ayana, 115 Angar, Ejere, and Lalistu)

• What is the sampling unit for this study? (Household or woman)?

• Inclusion criteria: Are women who reside in the area for less than six months included?

• Explanatory variables: The description regarding the Andersen Newman’s model (conceptual model) could be summarised and moved to the introduction, and focus on the list of predictor variables for this particular study.

• Please move or remove this “None of the participants refused to be interviewed. Five women wanted to end the interviews early due to personal appointments they had to attend to; they were reported as non-responders” to the appropriate section

Result

• Marital status (n=383) Vs Husband’s education and occupation (n=454)??

• Moslem or Muslim?

• Description at kebele, “We found that 89.4% and 66.2% of respondents of Ayana and Ejere Kebeles, respectively, visited the ANC clinic, and lower proportions did so in Angar and Lalistu Kebeles.” may not be required.

• Factors associated with ANC: The bivariate analysis could be merged with the multivariate model in one table to show how the variables are transferred from the first model to the next. Or else, authors can summarise the bivariate analysis in one paragraph without a table and then directly move to describe the analysis results of the final model.

• What is the (n=??) for the regression model?

• I’m not sure why authors consider the kebeles in the regression, as these might have insignificant differences in terms of socio-economic and demographic factors.

Discussion

• The discussion was nicely presented except the issues raised below

• You don’t need to refer to Table 4 in the discussion. Move Table 4 from this section

• The magnitude of use of the ‘minimum number of ANC’ was not discussed.

• Again, the factors do not need to be categorized here in the discussion. First, discuss the magnitude, and then focus on the modifiable factors associated with the utilization of the minimum number of ANC visits.

Reviewer #2: Dear Prof Kannan Navaneetham

Thank you for inviting me to review this interesting paper. I have a very minor collection of comments and hoping my comments are very easy for the author to make correction and I recommend you to accept the paper without modification. The author should also check my comments within the PDf version which I attached it with the word version of my comments. I look forward your invitation again when you have an MS that that be should be reviewed very carefully.

Thanking you.

Here are my minor comments which will improve the paper

Title

Q1. Title: Using abbreviation in title ANC alone is confusing, please write both the expansion and the abbreviations?

Q2. The short title is not actually short: I recommend this, Health care utilization and antenatal care Services in rural western Ethiopia.

Abstract

Q3. Abstract: Re-write the methods and result section separately for clarity to readers, please see my comments within the PDF

Q4. From the abstract result section, for all AOR, please include the confidence interval to see how the association is strong or weak? This is very key information for the scientific community.

Q5. From conclusion of the abstract, utilization of recommended ANC services was strongly linked with predisposing, enabling, need, and external factors? This seems very general better to mention those factors which are actually associated with among predisposing, enabling, need, and external factors due to all factors of predisposing, enabling, need, and external factors are not actually a factor for the use of ANC?

Q6. From line 34, which intervention you recommended as a researcher?

Q7. Key word is not applicable for PLoS ONE publication, thus, delete it.

Introduction

Q8. Referencing style should be in line with the PLoS OEN guideline? This comment for all throughout your MS.

Q9. Line 46-50 should be replaced by the current HSTP (Health Sector Development Plan) and SDG goals, why you sued the expired MDG and HSDP information?

Q10. Better to delete lines 81-83 since it is repetition with line 93-94. Keeping line 93-94 idea is good.

Methodology

Q11. Better to put the map of the study area, if the author put the study area in another published of his paper, he can cite that paper without putting the map within this paper. This gives a clear view for the readers of this paper.

Q12. Line 100 and line 102 refrence should be corrected. Use CSA reference for line 102 AND LINE 100 for Eastern Wollega Zone, Wone Finance and Economic Development Office. Physical and

561 Socio Economic Profile of Gidda Ayana Woreda. Nekemte, Ethiopia; 2015. Making both at the same time seems both data available in both sources, which could not be in real situation?

Q13. Line 104, better to write a ‘community based’ than population based. Again, the langue is not clear, Please see my comment within the PDF for line 104.

Q14. Line 110 to 112 had confusing ideas, if the source population less than 10,000, sample size correction is done and adding 5% non-response rate does not the justification. Adding a non-response rate either less than or greater than 10,000 is a must. Please re-write it/

Q15. Line 121 of using recent births, why not you select randomly one of the mother?

Q16. Line 125-126 seems it will incur bias? Btter to delete it.

Q17. Line 131. Cite the WHO recommendation source for ANC categorization?

Q18. Line 136 to 162 should not be one paragraph? Make it at-least 3

Q19. Line 165-66, re-write, What does it mean literature? Did you mean published paper

Q20, the full name and abbreviation for health extension workers written wt line 117, then at line 170 use only the abbreviations, such comment works for all. Once you used both the full name and the abbreviation, then use the abbreviation, check also about EDHS, ANC and etc.

Q21. Line 181 and 812, re-write for clarity.

Q22 Line 178-193, make it at least two paragraph?

Q23. Line 194. consent is part of ethics and no need to write as a topics , make it the title ‘Ethical consideration”

Result

Q24. Line 219, The exchange rate should be during your data collection period? And please write the time from your result display or in the table?

Q 25. Line 220, Table 1 should have information about mean age, and mean number of children women gave birth? Putting by text from line 211 is not enough?

Q26. Table 1 topic needs modification. There is no cultural related data in the table so that saying Socio-cultural and demographic backgrounds not correct? But the Table also have HEWs information then, you can find out better title as reflecting what exist in the Table?

Q27. For all Tables write the study period and the correct location of the study area by including rural western Ethiopia.

Q28. Liens 236-238 should be deleted since the bivariate association has no value? But I understand that Table 3 about multivariable analysis result and if so, the Table 3 title saying bivariate is total wrong and confusing? Please check

Q29. If Table 3 about Bivaraite analysis, your note on 235 to 307 should be deleted since bivariate data is noting for decision making and the association have no value.

Discussion

Q30. Please bring Table 4 above the discussion at result section?

Q31. Discussion better to be re-written? Repeating the result section at the discussion is not recommended, either you use result and discussion within the same topic or separately, I thought you have a separate topic for discussion, so in your discussion section focus on discussions part? No need to write each result again? At the result section, put all your result and focus on the main findings, you wasted your time at the result by bivariate result at Table 4, but the most important finding is Table 4.

Q31. Line 451, write the reason why no grant number

Q 32, 455, in your acknowledgement, please acknowledge also data collectors supervisors, Addis Abbaa University, Department f Geography at AAU and etc. This tells how you are very careful

6. PLOS authors have the option to publish the peer review history of their article (what does this mean?). If published, this will include your full peer review and any attached files.

Reviewer #1: No

Reviewer #2: Yes: Metadel Adane (PhD)

Assistant Professor of Water and Public Health

Department of Environmental Health

Wollo University

Dessie,. Ethiopia

---

## [Author Response · Author response to Decision Letter 0]

31 Dec 2019

Resubmission date: December 24, 2019

Manuscript ID: PONE-D-19-25468

Title: “Using Andersen’s behavioural model of health care utilization in a decentralized program to examine the use of antenatal care in rural western Ethiopia”

Authors: Habtamu Tolera, Tegegne Gebre-Egziabher, Helmut Kloos

Dear Dr. Navaneetham,

Thank you for your letter dated December 11, 2019. We were pleased to know that our manuscript was considered potentially acceptable for publication in PLOS ONE, subject to adequate revision as requested by the reviewers. Based on the instructions provided in your letter, we uploaded the file of the rebuttal letter; the marked-up copy of the revised manuscript highlighting the changes made in the original submitted version and the clean copy of the revised manuscript. 

We have revised the manuscript by modifying the abstract, introduction, methods, results, discussion and other sections, based on the comments made by the reviewers and using the journal guidelines. Accordingly, we have marked in red color all the changes made during the revision process. Appended to this letter is our point-by-point response to the comments made by the editor and the two reviewers. 

We agreed with almost all the comments/questions raised by the editor and the reviewers and provided justification for disagreeing with some of them. We would like to take this opportunity to express our thanks to the editor and the reviewers for their valuable comments and to thank you for allowing us to resubmit a revision of the manuscript. 

I hope that the revised manuscript is accepted for publication in PLOS ONE. 

Sincerely yours 

Habtamu Tolera

PhD Candidate in Socio-Economic Development in Addis Ababa University, Ethiopia 

Lecturer and Researcher in Wollega University, Ethiopia 

Phone: +251 (0) 912015545

E-mail address: habtol@yahoo.com

Response to the academic Editor’s and the reviewers’s comments

Response to the Editor-in-Chief’s comments

General comment: 

DearMr.Tolera,

Thank you for submitting your manuscript to PLOS ONE. After careful consideration, we feel that it has merit but does not fully meet PLOS ONE’s publication criteria as it currently stands. Therefore, we invite you to submit a revised version of the manuscript that addresses the points raised during the review process. We would appreciate receiving your revised manuscript by Jan 25 2020 11:59PM. To enhance the reproducibility of your results, we recommend that if applicable you deposit your laboratory protocols in protocols.io, where a protocol can be assigned its own identifier (DOI) such that it can be cited independently in the future. For instructions see: http://journals.plos.org/plosone/s/submission-guidelines#loc-laboratory-protocols

• A rebuttal letter that responds to each point raised by the academic editor and reviewer(s). This letter should be uploaded as separate file and labeled 'Response to Reviewers'.

• A marked-up copy of your manuscript that highlights changes made to the original version. This file should be uploaded as separate file and labeled 'Revised Manuscript with Track Changes'.

• An unmarked version of your revised paper without tracked changes. This file should be uploaded as separate file and labeled 'Manuscript'.

• We look forward to receiving your revised manuscript.

Kind regards,

Kannan Navaneetham

Academic Editor

PLOS ONE

Dear Editor, thank you for your positive decision and for your letter allowing us resubmitting our revised manuscript for publication. We all agreed, reconsidered and revised all concerns and commented raised in a point-by-point manner as you and your two reviewers rightly suggested. We addressed all points alternatively in the following sections. Thank once again!! 

1. Journal Requirements:

1.1. Please ensure that your manuscript meets PLOS ONE's style requirements, including those for file naming. The PLOS ONE style templates can be found at http://www.plosone.org/attachments/PLOSOne_formatting_sample_main_body.pdf and http://www.plosone.org/attachments/PLOSOne_formatting_sample_title_authors_affiliations.pd.

Thank you for your remark. We really appreciate your comments. We have read journal’s author guidelines and tried to meet PLOS ONE's citation style and formatting requirements, including those for file naming as academic editor suggested.

1.2. Please ensure that you refer to Figure 2 in your text as, if accepted, production will need this reference to link the reader to the figure.

Thank you for your comment. We refer to Figure 2 in the original submission as supportive information, S1_Fig.Doxc file, in the revised submission. You can check a list on page 34 lines 686 under Supportive information” statement.

1.3. Please include additional information regarding the survey or questionnaire used in the study and ensure that you have provided sufficient details that others could replicate the analyses. For instance, if you developed a questionnaire as part of this study and it is not under a copyright more restrictive than CC-BY, please include a copy, in both the original language and English, as Supporting Information. If the original language is written in non-Latin characters, for example Amharic, Chinese, or Korean, please use a file format that ensures these characters are visible.

Thank so much for your observation. We have uploaded the minimum anonymized data set as a supporting information files as suggested which include English language survey questionnaire (S1_text.docx), see the revised manuscript of “Supportive information" statement on page 34 lines 687-689.and Original Andersen Behavioural model for healthcare seeking (S1_Fig.Doxc). 

1.4. Please state whether you validated the questionnaire prior to testing on study participants. Please provide details regarding the validation group within the methods section.

Thank so much for your observation. As mentioned above, we reconsidered your concern and discussed in details under sub-section” Data collection and quality control” in this revised report. See page 7-8, lines 170-180. 

1.5. Please change your reference to "p=0.000" to "p<0.001" or as similarly appropriate, as p values cannot equal zero 

Thank so much for your observation and we revised as you fairly commented us. See Table 3 on pages 17-18, lines 336-337. 

Responses to Reviewer comments # 1 

1. Title: 

1.1. Please avoid the use of abbreviations like (ANC) in the title.

Thank you for your remark. We have agreed with your concern and taken out this acronym, “ANC”, from the study’s title as commented. See page 1, line 3.

2. Abstract

2.1. In the result part, it is better to describe the magnitude (15.2%) and then the factors. “Adherence to the recommended number of ANC service visits was a function of predisposing, enabling, need, and external environmental factors. Women who made the recommended number of ANC visits constituted 15.2% of all subjects”

First of all, we appreciate your constructive comment. We observed your concern much because it reshaped and further strengthen the value of the manuscript, and well-kept the flows of ideas. We have reworked accordingly. See page 1, lines 19-27 of the ‘Results’ section under Abstract.

2.2. Does this statement “Women’s involvement in gainful activities had higher log odds of seeking the recommended ANC services compared to housewives.” based on the logistic regression model result? If so, you should write the Odds ratio form the regression

Thank you so much for your comment. Hence, we incorporated the remark as suggested above. See page 1, lines 23-24 of the ‘Results’ section under Abstract.

3. Introduction

3.1. The statistics reported in the first paragraph, “About 300,000 maternal deaths were reported worldwide in 2013”, is rather reflected in the 2015 WHO maternal mortality report. Would you please double-check the figure?

Thank you for your remark. As per of your concern, different global figures in different publications or reports are available there online. We double checked and used recent report as you commented us and incorporated global figures reflected in the 2015 WHO maternal mortality report: Trends maternal mortality. 1990 to 2015: Estimates by WHO, UNICEF, UNFPA, World Bank Group and the United Nations Population Division. Geneva: World Health Organization.; 2015. See pages 1-2, lines 36-40 under the ‘Background’ section.

3.2. Paragraph two: “The Ethiopian Demographic Health Surveys of 2005 and 2011 revealed maternal death ratios of 673 and 676 per 100,000 live births, respectively, showing no change between the two studies.” The maternal mortality definition used by the DHS is different from the formal WHO definition used as it considers all pregnancy-related deaths. Hence using the figure from the DHS might not be correct in this context.

We appreciate your suggestion. As commented above, we have taken out local DHS figures we cited in the original submission and used WHO global figures on country’s report in place as per of your suggestion. See page 2, lines 41-43 under the ‘Background’ section. 

3.3. Please summarize the last three paragraphs of the introduction in one succinct sentence that addresses the justification and aim of the study.

Thank so much for your observation. Yes, we have understood your concern and merged the last three paragraphs of the original submission and modified to a concise paragraph as you indicated above in this revised submission. See pages 3-4, lines 69-93 under the ‘Background’ section. 

4. Method

4.1. Study setting: Add some brief information regarding the health infrastructure of the study area. 

We incorporated this concern as rightly indicated above. See page 4, lines 102-106 under the ‘Study setting’ section. 

4.2. You don’t need to mention the name of the kebeles “(Ayana, 115 Angar, Ejere, and Lalistu).

Alright, as a reviewer indicated in his report, we have taken out the name of the kebeles from ‘Sample and recruitment’ section under the ‘Methodology. See page 5, lines 119.

4.3. What is the sampling unit for this study? (Household or woman)?

We appreciate your observation. We used ‘Household’ or ‘woman’ interchangeably unintentionally. However, the actual case, sampling unit is the pregnant women not the household. Thus, modification has been made accordingly where it appears across the text in this revised submission. See page 5, lines 120-122, under this revised submission. 

4.4. Inclusion criteria: Are women who reside in the area for less than six months included?

Thank you for reminding us to mention. Women who came from somewhere else, outside the study woreda and did not reside in the study kebeles 6 months or fewer preceding data collection were not included in the study. This is indicated in the revised submission. See page 6, lines 129-130, under the sub-section ‘Sample and recruitment’. 

4.5. Explanatory variables: The description regarding the Andersen Newman’s model (conceptual model) could be summarized and moved to the introduction, and focus on the list of predictor variables for this particular study.

As marked above, we have summarized description about Andersen Model. See in the “Background” section on page 3-4, lines 78-88. See also the revised list of explanatory variables on pages 6-7, from lines 140-160 in the sub-section ‘Explanatory variables’ under the ‘Methodology’

4.6. Please move or remove this “None of the participants refused to be interviewed. Five women wanted to end the interviews early due to personal appointments they had to attend to; they were reported as non-responders to the appropriate section.

Thank you so much. We have taken out the above texts as per of your comment.

5. Result

5.1. Marital status (n=383) Vs Husband’s education and occupation (n=454)??

We checked everything about the concerns. we have checked but not observed no any arithmetical errors with regard to, n or N, in Table 1 or may we not understand a reviewer’s report?

5.2. Moslem or Muslim?

Thank so much for your comment on our inconsistency in not using one of the above across the manuscript. We have used “Muslim” to keep consistency throughout the whole revised submission. See, e.g., page 14, line 263 and elsewhere.

5.3. Description at kebele, “We found that 89.4% and 66.2% of respondents of Ayana and Ejere Kebeles, respectively, visited the ANC clinic, and lower proportions did so in Angar and Lalistu Kebeles.” may not be required.

Thank you so much for your observation. We all authors needed to see some variations observed in proportions of ANC service utilization among pregnant women across kebeles in which decentralized public primary health facilities were located. So kebeles are administrative decentralization entities in which health facilities or maternal programs including health workers were transferred one step down or are decentralization proxy variable. But mainly and primarily, we practically observed that there are some variations in cultural practice among study kebeles because Ayana and Lalistu kebeles are almost Oromo dominated host communities while Angar and Ejere are Amhara and Tigre dominated, originally settlers, from other regions outside Oromia region. This’s is our primary interest to analyse the frequency distribution of ANC utilization using kebeles as a categorical variable and the reason we also included kebeles in our regression model. See Table 2 on pages 12-13, and pages 12, lines 238-240 sub-section “ANC service use and decentralized facility attended”; see also the ‘External environmental factors’ part under ‘Result’ on page 17, from lines 325-329.

5.4. Factors associated with ANC: The bivariate analysis could be merged with the multivariate model in one table to show how the variables are transferred from the first model to the next. Or else, authors can summarise the bivariate analysis in one paragraph without a table and then directly move to describe the analysis results of the final model

Yes, as indicated above, bivariate analyses outputs are crude and are not as such important for decisions. Hence, we have summarised the data in the bivariate Table 3, in old submission in one paragraph in the new submission, see page 13, lines 249-258 and finally we deleted a Bivariate Table 3 of the original submission. 

5.5. What is the (n=??) for the regression model?

We appreciate your remark. As indicated above, n=?? is deleted in the new submission because it is our, an excused technical error. See Table 3, on page 17, line 336.

5.6. I’m not sure why authors consider the kebeles in the regression, as these might have insignificant differences in terms of socio-economic and demographic factors.

Thank, we appreciate your observation. This question is a part of questions raised in subsection 5.3 and it is well answered there. See above, subsection 5.3. 

6. Discussion: the discussion was nicely presented except the issues raised below

6.1. You don’t need to refer to Table 4 in the discussion. Move Table 4 from this section

We appreciate your suggestion. and we have incorporated both concerns. See the moved Table 3 on pages 17-18, lines 336-337 under the Result section in this revised submission, we also deleted unnecessary refer to Table 4 in the old submission under “Discussion” part.

6.2. The magnitude of use of the ‘minimum number of ANC’ was not discussed.

Thank you so much for constructive comment. As suggested, we have tried to satisfy a reviewer’s concern. We have given emphasis in the revised “Discussion” part of the new submission for the magnitude of use of the ‘minimum number of ANC visits as much as possible. See pages 18-24, lines 337-460.

6.3. Again, the factors do not need to be categorized here in the discussion. First, discuss the magnitude, and then focus on the modifiable factors associated with the utilization of the minimum number of ANC visits.

Thank you very much also. Authors have also taken out the categories used in the 

“Discussion” section of the original submission. See the revised version on pages 

19-24, lines 350-463.

Responses to Reviewer comments # 2 , Dr. Metadel Adane

General Comments:

Thank you for inviting me to review this interesting paper. I have a very minor collection of comments and hoping my comments are very easy for the author to make correction and I recommend you to accept the paper without modification. The author should also check my comments within the PDf version which I attached it with the word version of my comments. I look forward your invitation again when you have an MS that that should be reviewed very carefully.

Thank you for your kind recognition of our efforts. We also appreciate the positive and invaluable assessment and feedbacks you offer us to add the value on this work. Thank you once also. In the following sections, we have tried to respond on reviewer #2’s comments, concerns.

Title

Q1. Title: Using abbreviation in title ANC alone is confusing, please write both the expansion and the abbreviations?

Thank you so much for your observation. Authors have agreed with your concern and taken out this acronym, “ANC” in the title as you fairly suggested so as to avoid confusion among future readers of the article, see page 1, line 3.

Q2. The short title is not actually short: I recommend this, Health care utilization and antenatal care services in rural western Ethiopia.

We appreciate your concern. We have incorporated the comment indicated above during our system-based online uploading to the journal in the revised submission.

 Abstract

Q3. Abstract: Re-write the methods and result section separately for clarity to readers, please see my comments within the PDF. 

We have satisfied your concern. We have put both the methods and result sections under Abstract separately in the revised submission. See a front page, lines 14-27.But there is a word limit, 300 words, to describe the details

Q4. From the abstract result section, for all AOR, please include the confidence interval to see how the association is strong or weak? This is very key information for the scientific community.

We appreciate your invaluable comment and incorporated both AOR and 95%CI in both the “Result” section under the “Abstract” and throughout the manuscript in the revised submission. E.g., see page 1, lines 20-27; pages 14-18, lines 266-337.

Q5. From conclusion of the abstract, utilization of recommended ANC services was strongly linked with predisposing, enabling, need, and external factors? This seems very general better to mention those factors which are actually associated with among predisposing, enabling, need, and external factors due to all factors of predisposing, enabling, need, and external factors are not actually a factor for the use of ANC?

Yes, positive remark. This is modified as per of the above suggestion. See page 1 lines 28-29 in the revised version of the conclusion under the Abstract.

Q6. From line 34, which intervention you recommended as a researcher?

Thank you so much for positive comment and it has been rewritten to satisfy reviewer’s concern as suggested above. See page 1 lines 30-34 in the sub-section ‘Conclusion’ under the ‘Abstract’ in this revised submission.

Q7. Key word is not applicable for PLoS ONE publication, thus, delete it.

We appreciate the comment, excuse our failure of not keeping PLoS ONE’s author guide or template. As noticed, we have removed this section. See the revised submission in the front page.

Introduction

Q8. Referencing style should be in line with the PLoS OEN guideline? This comment for all throughout your MS.

Thank you, yes, we have to satisfy the requirements of the journal. We have corrected in-text citation style problem indicated above as per of PLoS OEN guideline, number in-text citations in rectangular bracket throughout the entire revised version. See the first in-text citation, as example on page 2 line 40. Thank you once again.

Q9. Line 46-50 should be replaced by the current HSTP (Health Sector Development Plan) and SDG goals, why you used the expired MDG and HSDP information?

It is a positive comment and we have modified accordingly. See page 2 from lines 41-43 in the “Introduction” section in the new submission. 

Q10. Better to delete lines 81-83 since it is repetition with line 93-94. Keeping line 93-94 idea is good.

We appreciate your comment. As indicated, we have removed the sentences from lines 81-83 in original submission and kept lines 93-94 . Of course, we almost modified the “Introduction” of the study to satisfy some concerns raised by reviewer # 1. See pages 2-4, lines 35-93 of the new submission.

Methodology

Q11. Better to put the map of the study area, if the author put the study area in another published of his paper, he can cite that paper without putting the map within this paper. This gives a clear view for the readers of this paper.

Thank you. As mentioned above, we have cited another work. See page 4 line 96 under a sub-section “Study setting”. 

Q12. Line 100 and line 102 reference should be corrected. Use CSA reference for line 102 AND LINE 100 for Eastern Wollega Zone, Wone Finance and Economic Development Office. Physical and 561 Socio Economic Profile of Gidda Ayana Woreda Nekemte, Ethiopia; 2015. Making both at the same time seems both data available in both sources, which could not be in real situation?

We accept the comment. We used both sources purposively because the 2007 Census release reported the total population as Gida-Kiremu Woreda. Hoever, after census the woreda was divided in to two woredas, Gida Ayana and Kiremu Woredas, and even same kebeles cross the border spilt into two, one side to Gida Ayana and the other side to Kiremu Woreda in 2008. This a challenge for us and we used both counts by the woredas and counts by the census. That’s why we cite both. So, this is the case and better if we cite both sources as they are originally indicated. See page 4 lines 99-102.

Q13. Line 104, better to write a ‘community based’ than population based. Again, the langue is not clear, please see my comment within the PDF for line 104.

We have incorporated the concern as per of the suggestion. It is also rewritten. See the revised version on page from 5 lines 108-109 in the sub-section “Study design and period” under Methodology part. 

Q14. Line 110 to 112 had confusing ideas, if the source population less than 10,000, sample size correction is done and adding 5% non-response rate does not the justification. Adding a non-response rate either less than or greater than 10,000 is a must. Please re-write it.

We have agreed with your concern and revised the issue as rightly suggested See page 5, lines 113-117 in the revised submission

Q15. Line 121 of using recent births, why not you select randomly one of the mothers?

We preferred to interview a mother with a recent birth if more than one housewife in a given household because it is more logical that this late mother recalls more of her reproductive history than that of her counterpart who gave births earlier than her in same household. Look at page 5 lines 120-122.

Q16. Line 125-126 seems it will incur bias? Better to delete it.

We have removed the entire sentence as a reviewer indicated above.

Q17. Line 131. Cite the WHO recommendation source for ANC categorization?

We have incorporated to satisfy the above concern. See page 6, from lines 134-35 in the sub-section “Outcome variable” under Methodology. 

Q18. Line 136 to 162 should not be one paragraph? Make it at-least 3

Aright, we have revised accordingly. Look at pages 6-7 from lines 141-160, in the new submission, a sub-section called “Explanatory variables”

Q19. Line 165-66, re-write, what does it mean literature? Did you mean published paper?

Thank you for the concern. As indicated, our interest is to mean it a published paper. Hence, we have rewritten as suggested. See page 7, line 164 in the revised version under a sub-section “Data collection and quality control”

Q20, the full name and abbreviation for health extension workers written at line 117, then at line 170 use only the abbreviations, such comment works for all. Once you used both the full name and the abbreviation, then use the abbreviation, check also about EDHS, ANC and etc.

We have appreciated you for the invaluable comment. We have incorporated the above specific comments and related suggestions across the manuscript. See e.g., page 7 line 168 in this revised version. 

Q21. Line 181 and 812, re-write for clarity.

Ok, revision is made as mentioned. See page 8 lines 185-187.

Q22 Line 178-193, make it at least two paragraphs?

Changes have been made as per of your suggestion. See pages 8-9 lines 182-203 of this revised version.

Q23. Line 194. consent is part of ethics and no need to write as a topic, make it the title ‘Ethical consideration”

We appreciate you for your observation. This is revised to satisfy the comment. See page 9 line 204. 

Result

Q24. Line 219, The exchange rate should be during your data collection period? And please write the time from your result display or in the table?

Truly, it is revised accordingly. See the foot note of Table 1 on page 12, line 233 in this revised submission.

Q 25. Line 220, Table 1 should have information about mean age, and mean number of children women gave birth? Putting by text from line 211 is not enough?

This is revised as suggested above. Look at page 10 lines 226-227.

Q26. Table 1 topic needs modification. There is no cultural related data in the table so that saying Socio-cultural and demographic backgrounds not correct? But the Table also have HEWs information then, you can find out better title as reflecting what exist in the Table?

We have revised incorporating all comments raised. See e.g., title of Table 1 page 10, lines 230-231; page 9 line 217, in the new submission.

Q27. For all Tables write the study period and the correct location of the study area by including rural western Ethiopia.

This is revised accordingly. See e.g., pages 10 lines 230-231.

Q28. Liens 236-238 should be deleted since the bivariate association has no value? But I understand that Table 3 about multivariable analysis result and if so, the Table 3 title saying bivariate is total wrong and confusing? Please check

We appreciate your concern. As per of your comment and also suggestions from the second reviewer we have deleted Table 3 or Bivariate Table. Rather, as reviewer #1 marked in his report as well we have preferred to summarizing Table 3 bivariate data into one concise paragraph immediately before proceeding discussion of multivariable analyses in the revised submission. See page13 lines

249-258. Finally, Table 3 was deleted.

Q29. If Table 3 about Bivariate analysis, your note on 235 to 307 should be deleted since bivariate data is noting for decision making and the association have no value

.

Thank you for your worry. Table 3 in the original version is as you said Bivariate Table. But the analysis preceding Table 3 or Bivariate Table is a Multivariable regression analysis. Thus, our note on lines 235 to 307 on the original submission should not be deleted since it is a multivariable regression analysis. We have moved Multivariable Table from the “Discussion” section to coincide the analysis section presented under “Results”. See pages 13-18 lines 259-337 in the revised submission. 

Discussion

Q30. Please bring Table 4 above the discussion at result section?

Alright, we have done it as commented and we modified the Table into Table 3. See pages 17-18 lines 336-337.

Q31. Discussion better to be re-written? Repeating the result section at the discussion is not recommended, either you use result and discussion within the same topic or separately, I thought you have a separate topic for discussion, so in your discussion section focus on discussions part? No need to write each result again? At the result section, put all your result and focus on the main findings, you wasted your time at the result by bivariate result at Table 4, but the most important finding is Table 4.

Thank you so much for your positive comments. We have revised the “Discussion” part as suggested. See pages 19-24 lines 350-463 in the revised submission.

Q31. Line 451, write the reason why no grant number.

Thank, the reason is that the amount of money offered is very small and is for direct data collection. It is difficult to label it as a full Ph.D. program fund and give a grant number. This is not the customary of the two institutions.

Q 32, 455, in your acknowledgement, please acknowledge also data collectors and supervisors, Addis Ababa University, Department of Geography at AAU and etc. This tells how you are very careful.

Thank, as you mentioned we acknowledged data collectors and the supervisors. See page 26 lines 506-507 in the new submission under ‘Acknowledgement” section. However, PLoS One author guideline restricts us not to include funding institution or competing interest’s information in Acknowledgments section. 

We would like to thank the reviewers and editors for evaluating our manuscript. We have tried to address all the concerns in a proper way and believe that our paper has been considerably improved. We would be happy to make further corrections if necessary and look forward to hearing from you all soon.

---

## [Decision Letter · Decision Letter 1]

13 Jan 2020

Using Andersen’s behavioral model of health care utilization in a decentralized program to examine the use of antenatal care in rural western Ethiopia

PONE-D-19-25468R1

Dear Dr. Tolera,

We are pleased to inform you that your manuscript has been judged scientifically suitable for publication and will be formally accepted for publication once it complies with all outstanding technical requirements.

With kind regards,

Kannan Navaneetham

Academic Editor

PLOS ONE

Additional Editor Comments (optional):

Reviewers' comments:

Reviewer's Responses to Questions

**Comments to the Author**

1. If the authors have adequately addressed your comments raised in a previous round of review and you feel that this manuscript is now acceptable for publication, you may indicate that here to bypass the “Comments to the Author” section, enter your conflict of interest statement in the “Confidential to Editor” section, and submit your "Accept" recommendation.

Reviewer #1: All comments have been addressed

Reviewer #2: All comments have been addressed

2. Is the manuscript technically sound, and do the data support the conclusions?

Reviewer #1: Yes

Reviewer #2: Yes

3. Has the statistical analysis been performed appropriately and rigorously? 

Reviewer #1: Yes

Reviewer #2: Yes

4. Have the authors made all data underlying the findings in their manuscript fully available?

Reviewer #1: Yes

Reviewer #2: Yes

5. Is the manuscript presented in an intelligible fashion and written in standard English?

Reviewer #1: Yes

Reviewer #2: Yes

6. Review Comments to the Author

Reviewer #1: (No Response)

Reviewer #2: The author is a smart researcher. He addressed every concern in scientific manner and I really appreciate his commitment.

7. PLOS authors have the option to publish the peer review history of their article (what does this mean?). If published, this will include your full peer review and any attached files.

Reviewer #1: No

Reviewer #2: Yes: Dr Metadel Adane (PhD in Water and Public Health).

Department of Environmental Health

Wollo University

Dessie, Ethiopia

---

## [Editor Report · Acceptance letter]

17 Jan 2020

PONE-D-19-25468R1 

Using Andersen’s behavioral model of health care utilization in a decentralized program to examine the use of antenatal care in rural western Ethiopia 

Dear Dr. Tolera:

I am pleased to inform you that your manuscript has been deemed suitable for publication in PLOS ONE. Congratulations! Your manuscript is now with our production department. 

With kind regards,

on behalf of

Professor Kannan Navaneetham 

Academic Editor

PLOS ONE